# No Association between Genetic Variants of the *COMT* and *OPRM1* Genes and Pain Perception among Patients Undergoing Total Hip or Knee Arthroplasty for Primary Osteoarthritis

**DOI:** 10.3390/genes13101775

**Published:** 2022-10-01

**Authors:** Alina Jurewicz, Andrzej Bohatyrewicz, Maciej Pawlak, Maciej Tarnowski, Mateusz Kurzawski, Anna Machoy-Mokrzyńska, Mariusz Kaczmarczyk, Anna Lubkowska, Monika Chudecka, Agnieszka Maciejewska-Skrendo, Katarzyna Leźnicka

**Affiliations:** 1Department of Orthopaedics, Traumatology and Musculoskeletal Oncology, Pomeranian Medical University, Unii Lubelskiej 1, 71-252 Szczecin, Poland; 2Department of Physiology and Biochemistry, Poznan University of Physical Education, Królowej Jadwigi St. 27/39, 61-871 Poznan, Poland; 3Department of Physiology in Health Sciences, Pomeranian Medical University in Szczecin, 54 Żołnierska St., 70-210 Szczecin, Poland; 4Institute of Physical Culture Sciences, University of Szczecin, 40b/6 Piastów St., 71-065 Szczecin, Poland; 5Department of Experimental and Clinical Pharmacology, Pomeranian Medical University in Szczecin, Al. Powstanców Wlkp.72, 70-111 Szczecin, Poland; 6Department of Clinical and Molecular Biochemistry, Pomeranian Medical University in Szczecin, Al. Powstanców Wlkp.72, 70-111 Szczecin, Poland; 7Department of Functional Diagnostics and Physical Medicine, Faculty of Health Sciences, Pomeranian Medical University, 54 Żołnierska St., 71-210 Szczecin, Poland; 8Faculty of Physical Education, Gdansk University of Physical Education and Sports, K. Górskiego St. 1, 80-336 Gdansk, Poland

**Keywords:** *COMT*, *OPRM1*, polymorphism, pain, osteoarthritis

## Abstract

Each year approximately 1 million total hip replacements are performed worldwide. The most common indications to choose this procedure are rest pain and pain after activity as well as functional limitations influencing daily activities. Experimental pain is highly variable by individuals, which is partly due to genetics. The aim of the study was to investigate a possible association of the catechol-O-methyltransferase (*COMT*) and μ-opioid receptor (*OPRM1*) genotypes with pain perception in patients undergoing total hip replacement and total knee replacement taking into account aspects such as age, sex and diabetes. The study included 207 patients (119 females, 88 males, median age 65 years, range 33–77) that qualified for surgical treatment (total hip replacement and knee arthroplasty) due to osteoarthritis. Pain sensitivity measurement was performed using a standard algometer. The genomic DNA was extracted from the buccal cells.. Single locus analysis was conducted using a general linear model. In the study group, we did not find statistically significant genetic associations between variants of *COMT* and *OPRM1* and pain thresholds/pain tolerance. The analysis of subjective pain perception using the visual analog scale did not show any relationship between the *OPRM1* rs1799971A>G variant and *COMT* rs4680, rs4633, rs4818 and rs6269.

## 1. Introduction

Osteoarthritis (OA) of the hip and knee, both in humans and animals, is an age-related disorder. Its earliest symptoms are pain and gait impairment (limp). The defects that occur in osteoarthritis are in many cases of genetic origin or the result of disorders such as abnormal joint structure, inadequate bone perfusion, fractures or traumatic joint sprains. Obesity and lifestyle problems such as dynamic overloads from sports activities or static loads from sedentary activities are also related to the pathomechanism of OA [1]. This disease manifests itself as joint pain, although the cause of pain in osteoarthritis is certainly multifactorial. Moreover, the joint pathology developed in animal models, especially acute rat arthritis models, does not fully match the pathophysiology of painful human arthritis [2].

Moreover, in patients with osteoarthritis, gradual limitation or lack of mobility is observed. Such effects are caused by painful joints leading to atrophy and muscle weakness or wasting and progressive painful muscle tension, swelling and stiffness of the joints. OA is progressive and, if left untreated, it leads to a significant reduction in mobility and even disability. Each year, approximately 1 million total hip replacements are performed worldwide, so the increasing number of such interventions in this and other joints have clear economic consequences [3,4].

It is known that subjective factors play a significant role in the perception of pain. On the other hand, individual differences between volunteers or patients in the groups tested were found in relation to clinical or experimental pain [5,6], especially in individuals with diabetes, in whom neuronal dysfunction led to a loss of pain sensation. [7]. Apart from subjective and environmental factors, there are a number of studies showing that modulation of pain perception depends on many genetic variants, with those related to catecholamine and opioid signaling being the most frequently tested due to their physiological function [8]. The results focusing on the association between these variants and individual differences in experimental pain in humans are inconclusive. Therefore, more research needs to be conducted to determine whether these genetic variants can explain individual differences, both in clinical and experimental pain [9,10,11].

For the last two decades the knock-down animal studies, twin studies, candidate gene approaches and genome-wide analyses have been widely used to study the genetic contribution to pain and its treatment in the adult population [12,13]. The most extensively studied and confirmed genetic variants are located within genes encoding the catechol-O-methyltransferase (*COMT*) and the μ-opioid receptor (*OPRM1*).

The subjective perception of pain is a multifactorial sensory phenomenon based on classical nociception. However, its intensity, like no other sensation, is modulated by the psychological aspect [14]. Individual variability in the perception of pain involves not only the differences pertaining to nociceptors and metabolizing enzymes, but also the intensity of the released transmitters. Thus, taking into account the genetic variability that determines such intracellular processes, it could be speculated that the intensity of subjectively perceived pain in people with various genotypes may differ from its actual strength. In such a case, the feeling of pain in those who experience it would be a consequence of some impairment or exhaustion of adaptive processes of genetically conditioned neuronal plasticity, that is a set of neurophysiological and neurochemical factors, which directly or indirectly contribute to the prevention or reduction of pain in most cases of tissue damage [15].

The aim of the present study was to investigate a possible association of the *COMT* and *OPRM1* genotypes with pain perception in patients undergoing total hip replacement (THR) and total knee replacement (TKR), taking into account aspects such as age, sex and diabetes.

## 2. Materials and Methods

### 2.1. Ethics Statement

The Bioethics Committee of the Regional Medical Chamber in Szczecin (Poland) approved the study (KB-0012/163/19). The investigation protocols were conducted ethically according to the World Medical Association Declaration of Helsinki and to the Strengthening the Reporting of Genetic Association studies statement (STREGA). The participants were informed about the purpose of the experiment and gave their written informed consent to participate in the study. All personal information and results were anonymous and were processed and stored in accordance with current regulations of data protection in Poland.

### 2.2. Participants

The study included 207 patients eligible for arthroplasty, 139 patients for THR and 68 patients for TKR due to severe primary OA. They were recruited based on their clinical condition and imaging studies evaluated by a single orthopedic surgeon during outpatient consultations at the orthopedic outpatient clinic of the First Independent Public Clinical Hospital in Szczecin, Poland. Of the 207 patients (119 females, 88 males, median age 65 years, range 33–77) in the study group, 162 (78.3%) were treated for diabetes, 155 for hypertension (74.9%) and 107 were obese (body mass index ≥ 30) (51.7%). Criteria for patient eligibility for surgery included pain and significant disability, as evidenced by limited range of motion and difficulty walking, confirmed by radiographic deformity (grades III and IV in the Kellgren and Lawrence classification of osteoarthritis). Secondary forms of OA such as rheumatoid arthritis, sequelae of avascular necrosis of the femoral head, posttraumatic arthritis and patients with probable or confirmed diabetic polyneuropathy fulfilling the criteria proposed by the Diabetic Neuropathy Expert Group, were excluded. All patients underwent hip or knee arthroplasty between 13 January and 29 September 2020. All examinations of the patients qualified for surgery were performed on the day of their admission to the hospital.

### 2.3. Pain Measurement

Two tests: Pressure pain threshold (PPT) and pressure pain tolerance (PTOL) were carried out to assess the response after mechanical stimulation using a standard FPN 200 algometer (Wagner Instruments, Greenwich, CT, USA) ranging from 0 to 20 kg, with an attached disc-shaped rubber tip of 1 cm^2^. Measurements were taken on both upper limbs on the back of the hand between the thumb and the forefinger. All the measurements were carried out by the same researcher in the morning hours. The participants were tested in a sitting position.

The participants were instructed on the application of the algometer and then were given an opportunity to use the device. All of them were informed about both the neurophysiological meaning of the test and the rules of behavior during its duration. After the participant manifested the sensed pain by saying “stop”, the test was continued until the subject could not stand the strength of the stimulus, which signified the end of the measurement. The first measurement was, thus, referred to as the pain threshold (PPT), and the second one was classified as tolerance to pain (PTOL).

After conducting the PPT and PTOL tests, the patients were asked to indicate the pain level on a scale from 0 to 10. The visual analog scale (VAS) was used to assess the degree of patients’ subjective pain in relation to the subjective amount of pain experienced by a given individual in life. The intensity of pain was assessed on a ten-point scale, where zero meant “No pain and discomfort” and 10 meant “The worst possible pain and discomfort”.

### 2.4. Genotyping

Genomic DNA was extracted from the buccal cells by a Genomic Micro AX SWAB Gravity (A&A Biotechnology, Poland) according to the producer’s protocol. All samples were genotyped using allelic discrimination assays with TaqMan^®^ probes (Applied Biosystems, Carlsbad, CA, USA) on a 7500 Fast Real-Time PCR Detection System (Applied Biosystems). In the *COMT1* gene, four SNPs were genotyped: C/T rs4633, A/G rs4680, C/G rs4818 and A/G rs6269, while in the *OPRM1* gene, one SNP was genotyped: A/G rs1799971. To discriminate the *OPRM1* rs1799971 alleles, TaqMan^®^ Pre-Designed SNP Genotyping Assays (Applied Biosystems, USA) (assay ID: C___8950074_1_), for *COMT1* rs4633, rs4680, rs4818 and rs6269 (assays ID: C___2538747_20, C__25746809_50, C___2538750_10 and C___2538746_1_, respectively) consisting of fluorescently labelled (FAM and VIC) minor groove binder (MGB) probes and two specific primers was used [16,17]. All samples were genotyped in duplicate.

### 2.5. Statistical Analysis

The Hardy–Weinberg equilibrium was tested using a fast exact test. Single locus analysis was conducted using a general linear model, assuming codominant (three genotypes), dominant and recessive models. Dominant and recessive models were constructed with respect to minor alleles. The regression of traits on haplotypes was conducted based on a generalized linear model (haplotype based analysis, allowing for ambiguous haplotypes). The effects of haplotypes were modeled as additive, dominant (heterozygotes and homozygotes for a given haplotype were assumed to have the same effects) and recessive (homozygotes of a given haplotype were assumed to have an effect on the trait). The HWE testing, single-locus and haplotype-based analyses were conducted using SNPassoc and haplo.stats R packages https://cran.r-project.org/ (accessed on 3 May 2022). A *p*-value < 0.05 was considered significant. We used false discovery rate to control the rate of type I error. FDR using the Benjamini–Hochberg method was applied separately for each SNP.

## 3. Results

The characteristics of individuals involved in the study, including the SNP genotypes are presented in Table 1. The group for clinical observation consisted of 119 women (57.5%) and 88 men (42.5%) aged 63.3 years ± 8.9 years. There were 162 patients (78.3%) treated for diabetes. The mean age of in the study group was 63.3 years (±8.9 years), respectively (Table 1).

The major allele frequencies and the probabilities for the Hardy–Weinberg hypotheses are shown in Table 2. The genotyping error was assessed as 1%, while the call rate (the proportion of samples in which the genotyping provided an unambiguous reading) exceeded 95%. The missingness for all SNPs genotyped in the *COMT* gene was assessed as 1.9% (the genotypes for rs4633, rs4680, rs4818 and rs6269 genotypes were not described for four participants). With respect to the *OPRM1* gene, the missingness was 2.4% (the genotypes for rs1799971 genotypes were not described for five participants). None of the polymorphisms deviated from expectations under the assumption of allele equilibrium in a population. The lowest minor allele frequency exceeded 10% (rs1799971). Gender identity was inferred from X chromosome heterozygosity. To identify the unexpected relatedness, the degree of recent shared ancestry (identity by descent, IBD) for every pair of individuals in a study was estimated using a method of moments procedure implemented with PLINK software [18].

Results of single locus analysis under the assumption of the general model, and dominant and recessive patterns of inheritance are presented in Appendix A for *COMT* and in Appendix A for *OPRM1* (Appendix A). The studied pain characteristics: pain threshold, pain tolerance and subjective pain assessment (VAS) were not associated with the studied polymorphisms of the *COMT* and *OPRM1* genes.

Five haplotypes were reconstructed using the expectation–maximization algorithm. Haplotypes and associated frequencies are shown in Table 3. General linear model (GLM) regression of the pain characteristic on *COMT* haplotype effects (additive, dominant and recessive), allowing for ambiguous haplotypes was performed (Table 4).

The haplotype [C;G;G;G] was associated with PPTHR under the additive and dominant model. PPTHR was 0.57 lower among individuals with one copy of the [C;G;G;G] haplotype (compared with homozygous for the [T;A;C;A] haplotype). The effect is doubled in individuals having two copies of the [T;A;C;A] haplotypes) (Table 4). The effect was −0.92 in the carriers of at least one [C;G;G;G] haplotype (dominant model). We found a similar, although, smaller effect of the [C;G;G;G] on the PPTVASHL under the dominant model. Other pain characteristics were not associated with *COMT* haplotypes.

## 4. Discussion

The dysfunction of the diseased hip or knee joint is transferred both up and down to the adjacent motion system components (spine and other joints of the lower limb) and leads to a reduction in the range of motion, disturbs the gait efficiency and, above all, causes pain. In clinical practice, we observe that pain acceptance is much more difficult than the restriction of mobility. In parallel, living with insufficiently controlled pain significantly worsens physical and mental functioning [19].

The search for the genetic aspects of pain has already allowed the selection of several candidate genes. *COMT* and *OPRM1* are considered to be potential “pain genes”, since their products are functionally associated with pain susceptibility and analgesia.

The μ-opioid receptor is a critical receptor for endogenous and exogenous opioids analgesic substances, such as β-endorphin, enkephalin and morphine; hence, it is of high importance in the physiological and psychological response to stress, trauma and pain [20]. The A118G (A>G functional substitution at locus 118; rs1799971) polymorphism is one of the most frequently investigated single-nucleotide polymorphism (SNPs) in the *OPRM1* gene. The variant has an effect on a putative glycosylation site and the protein stability of the μ-opioid receptor, and also, reduces receptor expression and receptor signaling efficiency [21,22]. It has been shown that the rs1799971 polymorphism is involved in the need for analgesia in chronic pain and in both pain perception and pain management has been shown [23,24,25].

Catechol-O-methyltransferase is the key enzyme involved in degradation of catecholamines, especially adrenaline, noradrenaline and dopamine, all of which are engaged in numerous psychological and physiological processes, including the modulation of pain [26]. Three common SNPs in the *COMT* gene—rs4633, rs4680 and rs4818—are located within the central coding region of both membranous and soluble forms of COMT (S-COMT and MB-COMT, respectively). The fourth SNP, rs6269 is located in the promoter region and together (a system of four SNPs in the *COMT* gene) they form a haplotype. Rs4633, rs6929 and rs4818 are synonymous, whereas variation in SNP rs4680 is nonsynonymous and leads to a substitution of valine to methionine at codon 158. This variation affects the enzyme activity, neurotransmitter levels and pain thresholds [27].

In our study, the relationship between the four *COMT* and *OPRM1* rs1799971 genotypes and the degree of pain sensation was analyzed by performing the PPT and PTOL assessments and a visual analog scale (VAS) test. In the group of 207 patients that qualified for surgical treatment (THR and TKR) due to osteoarthritis, no statistically significant differences between the *COMT* and *OPRM1* genotype distributions and the pain threshold and tolerance were found. No statistically significant differences were, also, observed between the *COMT* and *OPRM1* polymorphisms studied and the subjective assessment of pain on the VAS scale. Furthermore, factors such as sex, age and diagnosed or (treated) diabetes did not influence the differences in the pain threshold and pain tolerance in patients with different variants of the *COMT* and *OPRM1* genes.

In turn, Zubiet et al. noticed [28] that *COMT* rs4680 AA homozygotes (Met/Met) were characterized by reduced regional μ-opioid system response to pain compared to AG heterozygotes (Met/Val) and GG homozygotes (Val/Val). Similar results were obtained by Martínez-Jauand et al. [29] in patients with fibromyalgia, who were characterized by a higher enzymatic activity of COMT compared to healthy people. Moreover, it was found that re4680 AA homozygotes experienced more experimental pain compared to GG, which would indicate an increased pain sensitivity.

Another research study carried out by Ahlers et al. [30] revealed that in patients with acute pain after cardiac surgery treated with intravenous infusions of morphine the rs4680 AA genotype in the *COMT* gene was related to a higher NRS score. One year later, this observation was not confirmed by Rut et al. [31] in patients suffering from chronic lumbar pain, as there were no significant associations between the *COMT* Val158Met polymorphism and the result on the VAS scale.

Rakvåg et al. [32] revealed that *COMT* rs4680 GG homozygous patients with cancer of different origin (breast, lung, abdominal cavity and urogenital system) received, on average, 50% higher daily doses of morphine compared with AA homozygotes. The association of the *COMT* rs4680: G>A gene polymorphism with opioid demand in patients following elective hip replacement was also assessed by Białecka et al. (2016). In this study, no relationship was found between the opioid requirement in the early postoperative period and the *COMT* rs4680: G>A polymorphism. These authors described unpublished data from their study based on *COMT* rs6269: A>G; rs4633: C>T; rs4818: C>G; SNP rs4680: G>A, and indicated the lack of significant relationships between the need for opioids and *COMT* rs4680: G>A SNPs as well as *COMT* haplotypes [33].

However, many different studies have shown that haplotypes composed of the *COMT* alleles of rs6269, rs4633, rs4818 and rs4680 alleles of the *COMT* gene influence the expression and activity of the COMT enzyme and correlate with pain responses [34,35]. The role of the haplotypes was further confirmed in other pain-related contexts [36,37]. In our study we noted that *COMT* haplotypes affect the pressure pain threshold as indicated in Table 4.

As shown above, the association of the most frequently studied *COMT* rs4680:G>A gene polymorphism with pain tolerance (treatment outcome with analgesics) is still debatable, as there is paucity of the available data, and the reports are not consistent. Treatment of chronic pain is becoming a global challenge, as it has a negative physiological, emotional and social impact on the dimensions of quality of life

Another SNP analyzed was the gene encoding the μ-opioid receptor (*OPRM1*). Our findings indicate no association between the *OPRM1* rs1799971A>G variant and the pain threshold, pain tolerance and subjective pain assessment (VAS). Different results to ours were obtained by Olesen et al. [38] who assessed the association of gene polymorphisms on the pressure pain threshold in patients with OA. Participants with the minor G allele of rs1799971 displayed higher pressure pain sensitivity. In turn, the results of a study conducted among cancer patients with the AA *OPRM1* genotype showed that they required significantly lower daily doses of morphine than patients with different genotypes due to the lower threshold of pain excitability [39].

In a study by Fillingim et al. [40], *OPRM1* genotyping showed that the carriers of the rare AII8G allele had significantly higher indices of the pressure–pain threshold than those with two alleles—homozygous—for the common allele. This allele occurred in 25% of the women and 17% of the men studied. The results indicate that this genotype may be associated with pain sensation in a gender-specific manner, and the rare G allele is associated with higher pressure pain thresholds.

## 5. Limitation

There were several limitations to this research. In the case of the genetic association study the relatively small group sample was the main limitation. Therefore, the research should be repeated within bigger, independent groups. Moreover, the genetic ancestry was not considered in the model in this study. However, assuming that the studied Polish population is genetically homogeneous, it can be assumed that the influence of this factor is unlikely to be significant. The predisposition to arthrosis is generally more noticeable in females, and it leads to asymmetric sex distribution (119 female and 88 male patients). A long duration of pain related to severe arthritis could create the change the mental acceptance of discomfort and pain and creates the abuse of painkillers. Because of this mechanism, we were forced, in some cases, to educate patients to more precisely estimate of the sensed force of the applied pain stimuli. Additionally, the experimental tests were carried out in a hospital setting in the preoperative period among distressed patients, which could have modified the reception of the pain stimuli.

## 6. Conclusions

The aim of this study was to determine a possible association between *COMT* and *OPRM1* genotypes and pain perception in patients undergoing total hip or total knee arthroplasty. The results obtained did not confirm this relationship in terms of pain threshold, pain tolerance and subjective pain rating on the VAS scale. To fully reject or confirm the hypothesis about the association between different variants of *COMT* and *OPRM1* genes and pain perception, further large-scale studies are needed to clarify the role of genetic factors in pain perception. Such observations may be used in the future to develop a program for individualized treatment of patients, particularly those with movement disorders.

## Figures and Tables

**Table 1 genes-13-01775-t001:** Demographic, clinical and genotypic characteristics of the individuals.

Characteristic	Mean ± SD, N (%)
Age (years)	63.3 ± 8.9
BMI (kg/m^2^)	30.3 ± 5.0
Sex (F/M)	119 (57.5)/88 (42.5)
Pain threshold (LH)	6.33 ± 2.40
Pain threshold (RH)	7.19 ± 2.61
Pain threshold (VASLH)	3.94 ± 1.74
Pain threshold (VASPH)	3.92 ± 1.82
Pain tolerance (HL)	10.87 ± 4.09
Pain tolerance (VASLH)	6.81 ± 1.73
Pain tolerance (RH)	11.28 ± 4.23
Pain tolerance (VASPH)	7.17 ± 1.58
Diabetes mellitus (Y/N)	162 (78.3)/45 (21.7)

LH—left hand, RH—right hand and VAS—visual analog scale.

**Table 2 genes-13-01775-t002:** Descriptive analysis of the catechol-O-methyltransferase (*COMT*) and μ-opioid receptor (*OPRM1*) variants.

	Minor Allele	Minor Allele Frequency (%)	HWE (*p*)	Missing (%)
*COMT* rs4633	C	47.3	0.206	1.9 (*n* = 4)
*COMT* rs4680	G	47.0	0.261	1.9 (*n* = 4)
*COMT* rs4818	G	35.5	0.539	1.9 (*n* = 4)
*COMT* rs6269	G	35.5	0.539	1.9 (*n* = 4)
*OPRM1* rs1799971	G	10.4	0.455	2.4 (*n* = 5)

**Table 3 genes-13-01775-t003:** Catechol-O-methyltransferase (*COMT*) haplotype frequencies.

Haplotype rs4633;rs4680;rs4818;rs6269	Frequency
[T;A;C;A]	0.52723
[C;G;G;G]	0.35149
[C;G;C;A]	0.11634
[C;A;C;G]	0.00248
[C;G;G;A]	0.00248

**Table 4 genes-13-01775-t004:** Regression of pain thresholds/tolerance on catechol-O-methyltransferase (*COMT*) haplotypes assuming an additive, dominant or recessive effect (adjusted for age and sex).

**Haplotype**	**Additive**							
	PPTLH	PPTRH	PPTVASLH	PPT VASRH	PTOLLH	PTOLRH	PTOLVASLH	PTOLVASRH
[C;G;C;A]	−0.18 (0.629)	−0.30 (0.457)	−0.16 (0.540)	−0.06 (0.835)	−0.37 (0.536)	−0.39 (0.536)	−0.11 (0.667)	−0.03 (0.903)
[C;G;G;G]	−0.14 (0.586)	−0.57 (0.047/0.768)	−0.36 (0.058)	−0.37 (0.065)	−0.04 (0.931)	0.04 (0.936)	−0.20 (0.292)	−0.21 (0.227)
[C;A;C;G] + [C;G;G;A]	−1.35 (0.420)	−0.80 (0.666)	−0.41 (0.739)	0.12 (0.923)	−2.55 (0.352)	−3.73 (0.197)	−1.35 (0.273)	−1.72 (0.119)
	**Dominant**							
	PPTLH	PPTRH	PPTVASLH	PPTVASRH	PTOLLH	PTOLRH	PTOLVASLH	PTOLVASRH
[C;G;C;A]	−0.10 (0.809)	−0.16 (0.718)	−0.14 (0.631)	−0.06 (0.838)	−0.25 (0.708)	−0.24 (0.734)	−0.09 (0.754)	−0.04 (0.886)
[C;G;G;G]	−0.32 (0.346)	−0.92 (0.014/0.768)	−0.50 (0.044/0.768)	−0.48 (0.072)	−0.28 (0.614)	−0.42 (0.482)	−0.24 (0.339)	−0.27 (0.230)
[C;A;C;G] + [C;G;G;A]	−1.24 (0.460)	−0.55 (0.764)	−0.30 (0.807)	0.21 (0.871)	−2.41 (0.380)	−3.51 (0.226)	−1.30 (0.291)	−1.68 (0.131)
	**Recessive**							
	PPPTLH	PPTRH	PPTVASLH	PPTVASRH	PTOLLH	PTOLRH	PTOLVASLH	PTOLVASRH
[C;G;C;A]	−1.19 (0.385)	−1.42 (0.348)	−0.01 (0.992)	0.63 (0.552)	−2.48 (0.265)	−3.19 (0.174)	−0.15 (0.883)	0.45 (0.620)
[C;G;G;G]	0.24 (0.647)	−0.11 (0.856)	−0.26 (0.505)	−0.41 (0.318)	0.67 (0.443)	1.32 (0.149)	−0.22 (0.566)	−0.20 (0.575)
[C;A;C;G] + [C;G;G;A]	NA	NA	NA	NA	NA	NA	NA	NA

Linear model coefficients and corresponding *p*-values in parentheses are shown: *COMT* haplotype [rs4633;rs4680;rs4818;rs6269] and reference haplotype [T;A;C;A]. LH—left hand, RH—right hand, VAS—visual analog scale; PPT: pressure pain threshold and PTOL: pressure pain tolerance. After slash—FDR adjusted *p* values.

## Data Availability

The data presented in this study are available on request from the corresponding author.

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
