# Peer review of "No Association between Genetic Variants of the COMT and OPRM1 Genes and Pain Perception among Patients Undergoing Total Hip or Knee Arthroplasty for Primary Osteoarthritis"

_genes, 2022, doi:10.3390/genes13101775_

Round 1

Reviewer 1 Report (New Reviewer)

This is a nice study evaluating the role of COMT and OPRM1 genetic variants on the perception of pain in patients qualifying for THR and TKR. The study is well written and structured. However, some methodological issues should be solved before considering publication:

- Were surgeries performed by the same surgeon or team? For patients undergoing THR: which surgical approach was used? For all patients: were the same implants used for both THR and TKR? Either yes or no, this information should be specified because it may introduce bias towards the perception of pain, as different surgical approaches and different implants (more or less invasive/constrained) can profoundly influence the outcomes. Please provide such details and summarize them in a Table.

- It is not clear when patients were tested. In lines 101-102, it is mentioned that "All examinations of the patients qualified for surgery were performed on the day of their admission to the hospital", but the aim of the study is, as per lines 77-80, "to investigate a potential association of the COMT and OPRM1 genotypes with pain perception in patients after total hip replacement (THR)  and total knee replacement (TKR)". When was pain measured? Before or after surgery? In the latter case, at what timepoint?

- Why was diabetes considered among the covariates? This has not been contextualized anywhere in the text.

Minor comments:

- Please format the Abstract as per MDPI's Instructions for Authors. Headings should be omitted.

- Introduction, lines 44-45: A more thorough introduction on osteoarthritis should be provided, with details regarding epidemiology and pathophysiology. Age is only one among many risk factors for OA. Gait impairment is not a frequent early symptom but may be related to chronic pain and development of joint deformity. Moreover, considering the aim of the study, a review of the main mechanisms behind osteoarthritic pain would be welcome.

Author Response

Dear Reviewer,

We would like to thank you for your careful reading of our manuscript and for your insightful comments, which helped us to improve our work considerably. We have corrected the manuscript and are addressing the comments.

Authors

 This is a nice study evaluating the role of COMT and OPRM1 genetic variants on the perception of pain in patients qualifying for THR and TKR. The study is well written and structured. However, some methodological issues should be solved before considering publication:

- Were surgeries performed by the same surgeon or team? For patients undergoing THR: which surgical approach was used? For all patients: were the same implants used for both THR and TKR? Either yes or no, this information should be specified because it may introduce bias towards the perception of pain, as different surgical approaches and different implants (more or less invasive/constrained) can profoundly influence the outcomes. Please provide such details and summarize them in a Table.

Authors: We have added the specified items to the description in the Materials and Methods, Participants section.

- It is not clear when patients were tested. In lines 101-102, it is mentioned that "All examinations of the patients qualified for surgery were performed on the day of their admission to the hospital", but the aim of the study is, as per lines 77-80, "to investigate a potential association of the COMT and OPRM1 genotypes with pain perception in patients after total hip replacement (THR)  and total knee replacement (TKR)". When was pain measured? Before or after surgery? In the latter case, at what timepoint? 

Authors: Thank you for bringing this inaccuracy to my attention. . All pain measurements were made by the same researcher in the morning hours of the day of admission, approximately 24 hours before surgery. We have changed the title of the paper accordingly and also improved the purpose of the study

- Why was diabetes considered among the covariates? This has not been contextualized anywhere in the text.

Authors:  Thank you for your attention to this issue. . We have supplemented the introduction with information on the perception of pain in patients with diabetes.

Minor comments:

- Please format the Abstract as per MDPI's Instructions for Authors. Headings should be omitted.

Authors:  Has been corrected

- Introduction, lines 44-45: A more thorough introduction on osteoarthritis should be provided, with details regarding epidemiology and pathophysiology. Age is only one among many risk factors for OA. Gait impairment is not a frequent early symptom but may be related to chronic pain and development of joint deformity. Moreover, considering the aim of the study, a review of the main mechanisms behind osteoarthritic pain would be welcome.

Authors:  As suggested, we have completed the description in the introduction

Reviewer 2 Report (New Reviewer)

Dear Authors: I want to congratulate with You for Your work.

In my opinion, some revision is needed:

- the title of the paper should be change in "No Association between Genetic Variants of the COMT and OPRM1 Genes and Pain Perception among Patients undergoing Total Hip or Knee Arthroplasty for Osteoarthritis"

- diagnosis was OA for all patients? I would write that "exclusion criteria" were rheumatoid arthritis, or I would write that diagnosis was "degenerative or inflammatory arthritis"; it would imply changes in the title as well

- possible biases can be 1) diagnosis, 3) assumption of pain-killer medicaments, 3) degree of degenerative OA, 4) comorbidities other than diabetes, .. You should better underline limitations 

best regards,

Author Response

Dear Reviewer,

We would like to thank you for Your careful reading of our manuscript and for Your comments, which helped us to improve our work considerably. We have corrected the manuscript and are addressing the comments.

Authors

Dear Authors: I want to congratulate with You for Your work.

In my opinion, some revision is needed:

- the title of the paper should be change in "No Association between Genetic Variants of the COMT and OPRM1 Genes and Pain Perception among Patients undergoing Total Hip or Knee Arthroplasty for Osteoarthritis"

Authors:  Thank you for bringing this inaccuracy to our attention. Of course, we have changed the title of the manuscript.

- diagnosis was OA for all patients? I would write that "exclusion criteria" were rheumatoid arthritis, or I would write that diagnosis was "degenerative or inflammatory arthritis"; it would imply changes in the title as well

- possible biases can be 1) diagnosis, 3) assumption of pain-killer medicaments, 3) degree of degenerative OA, 4) comorbidities other than diabetes, .. You should better underline limitations 

Authors:  We thank you for your comments. We have supplemented and corrected the given points according to your suggestions.

Reviewer 3 Report (New Reviewer)

Major corrections:

Could you please justify the use buccal cells (from the cheeks of patients?) for analysis of a disease associated with bone/musculoskeletal system? Why was this study not performed for data using bone cells like osteoblasts or stem cells or even blood samples from OA patients?

Could you please add more details to indicate how these buccal cell samples were selected?

Minor suggestions:

1.     Please add median age range, number of males and females in he patient cohort in the abstract

2.     Please add the same details in more details in section 2.2 – Participants. If you have other criteria available for the patients like what drugs they have been on etc – then consider adding those too as a paragraph.

3.     Please provide a reference for stating that the methods used in section 2.3 are standardised methods as evidence for readers.

4.     Similarly, please provide evidence for sections 2.4 and 2.5 so that readers know that your method are optimised and well-established.

Author Response

Dear Reviewer,

We would like to thank you for your careful reading of our manuscript and for your insightful comments, which helped us to improve our work considerably. We have corrected the manuscript and are addressing the comments.

Authors

Could you please justify the use buccal cells (from the cheeks of patients?) for analysis of a disease associated with bone/musculoskeletal system? Why was this study not performed for data using bone cells like osteoblasts or stem cells or even blood samples from OA patients?

Authors:  In the present study, association studies were performed on the occurrence of specific polymorphic variants of a single nucleotide polymorphisms in the genomic DNA of people with a specific disease entity. The genomic DNA sequence is unchanged in all somatic cells of the tested individual, so it does not matter whether we genotype the presence of a specific SNP variant from the buccal epithelial cells or the oseoblast, because the genotype identified for a given individual will be the same regardless of which somatic cell it originates from. The use of buccal epithelial cells (usually described as buccal swabs or cheek swabs) is a common standard and routine technique used in human genomic DNA studies because it is a noninvasive method, biological material is easy to obtain, and subsequent procedures for isolating genomic DNA from epithelial cells are simple

Gilbert JR, Vance JM. Isolation of genomic DNA from mammalian cells. Curr Protoc Hum Genet. 2001 May;Appendix 3:Appendix 3B. doi: 10.1002/0471142905.hga03bs19.;

Erickson SW, MacLeod SL, Hobbs CA. Cheek swabs, SNP chips, and CNVs: assessing the quality of copy number variant calls generated with subject-collected mail-in buccal brush DNA samples on a high-density genotyping microarray. BMC Med Genet. 2012 Jun 26;13:51. doi: 10.1186/1471-2350-13-51.

Could you please add more details to indicate how these buccal cell samples were selected?

Authors: The buccal cells donated by the participants were collected in 200 µl sterile water with the use of sterile foamtipped FLOQSwabs® applicators (COPAN Diagnostics Inc., USA). DNA was extracted from the buccal cells using a Genomic Micro AX SWAB Gravity (A&A Biotechnology, Poland) according to the manufacturer’s protocol.

 Minor suggestions:

  1. Please add median age range, number of males and females in he patient cohort in the abstract.

Authors:  The abstract has been completed.

  1. Please add the same details in more details in section 2.2 – Participants. If you have other criteria available for the patients like what drugs they have been on etc – then consider adding those too as a paragraph.

Authors:  We have completed the sections 2.2.- Participants with the variables available to us.

  1. Please provide a reference for stating that the methods used in section 2.3 are standardised methods as evidence for readers.

Authors:  Both PPT and PTOL using an algometer and the VAS as a subjective pain rating scale have been established measurement methods1 for decades and are used in studies published in about 400 papers per year. Therefore, we also used the methods that allowed us to compare and relate our own results to studies conducted on different groups of healthy, ill or suffering people of all ages and both sexes

1Delgado DA, Lambert BS, Boutris N, McCulloch PC, Robbins AB, Moreno MR, Harris JD. Validation of digital visual analog scale pain scoring with a traditional paper-based visual analog scale in adults. Journal of the American Academy of Orthopaedic Surgeons. Global research & reviews. 2018 Mar;2(3).

  1. Similarly, please provide evidence for sections 2.4 and 2.5 so that readers know that your method are optimised and well-established.

Authors:  We have added  the appropriate items according to your suggestions.

Round 2

Reviewer 1 Report (New Reviewer)

The authors have significantly increased the quality of their paper according to reviewers' comments. However, some other points should be regarded:

- Line 50: Defining OA pathophysiology as "largely genetic" is not adequate, as genetic influences are outweighed by additional risk factors of different nature.

- Please be consistent in the use of abbreviations throughout the text after mentioning them the first time.

- Line 67: It is better to refer to individuals with diabetes not as diabetics.

- Lines 67-68: The influence of diabetes in this population is still not clear. Alterations of nociception are certainly part of diabetic neuropathy, whose main presentation is diabetic polyneuropathy mostly affecting hands and feet. However, other less common forms of neuropathy also exist and these may also affect larger joints such as the knee and the hip. Has diabetic neurological dysfunction been evaluated in the study population? If no, investigating this subpopulation is not meaningful for the study.

- It is still not clear when patients were evaluated. If authors means the morning hours before operation (definitely preop), there is no need to provide any information about the surgery (including techniques and implants), which was not clear before.

Author Response

Dear Reviewer,

Thank you for pointing out the inaccuracies in the text. We have corrected all the indicated fragments in the manuscript.

Kind regards,

Authors

Reviewer 2 Report (New Reviewer)

Dear Authors:

Thank You for the corrections. I'm fine with the paper in the present form. best regards, and congrats.

Author Response

Thank You. 

Authors

Reviewer 3 Report (New Reviewer)

NA 

Author Response

Thank You.

Authors

This manuscript is a resubmission of an earlier submission. The following is a list of the peer review reports and author responses from that submission.

Round 1

Reviewer 1 Report

The main problem of this manuscript is in study design. I have two main complaints. 

First of all, your thesis is that there are an association between COMT and OPRM1 genotypes and pain perception in patients with osteoarthritis of the hip and knee. Honestly, you have proved some connections between them, but the main problem is in your choice of participants group. You have 207 patients with osteoarthritis, you did them two tests and VAS assesment and you got some results. My question is: Why did you decide to include patients with osteoarthritis? In study like this, instead of the patients with osteoarthritis you could include what ever group of people you want  and you will get the same or similar results. If you wanted to have o focus on a group of patients with the same (or similar) diagnosis, you should have a control group of healthy people (or persons without the same diagnosis). Only in that case you will be able to compare results and your conclusion will have a scientific significance. You wrote in line 112 and 113 "A long duration of pain related to severe arthritis could change the mental acceptance of discomfort and pain and because of this mechanism influence the reception of the applied pain stimuli". I totally agree. One more reason to have healthy persons in control group.

Secondly, all 207 patients were in the same group. Osteoarthritis has some stages, there are a lot of classifications, maybe the most popular is Kellgren--Lawrence scale. How can you be sure that there are no differences between patients who are in a different stage (or grade) of the disease? You should divide your participants in a few groups, according the stage of the disease. After measurements, you should compare results between subgroups, according to the stage or level of the osteoarthritis.

I hope my comments will help you to improve your manuscript.

Author Response

Dear Reviewer,

We would like to thank for careful reading our manuscript and for giving insightful comments, which helped us significantly improve our paper. We have revised the manuscript and respond to the comments of the reviewer.

The Authors.

The main problem of this manuscript is in study design. I have two main complaints.First of all, your thesis is that there are an association between COMT and OPRM1 genotypes and pain perception in patients with osteoarthritis of the hip and knee. Honestly, you have proved some connections between them, but the main problem is in your choice of participants group. You have 207 patients with osteoarthritis, you did them two tests and VAS assesment and you got some results. My question is: Why did you decide to include patients with osteoarthritis? In study like this, instead of the patients with osteoarthritis you could include what ever group of people you want  and you will get the same or similar results. If you wanted to have o focus on a group of patients with the same (or similar) diagnosis, you should have a control group of healthy people (or persons without the same diagnosis). Only in that case you will be able to compare results and your conclusion will have a scientific significance. You wrote in line 112 and 113 "A long duration of pain related to severe arthritis could change the mental acceptance of discomfort and pain and because of this mechanism influence the reception of the applied pain stimuli". I totally agree. One more reason to have healthy persons in control group.

Authors:
The purpose of this research was to describe both the relationship between COMT and OPRM1 genotypes and as well as pain sensitivity in patients with osteoarthritis of the hip and knee. The reviewer rightly noted that this goal through  too generally formulation, may raise some doubts. It is worth noting, however, that in this research project, also additional variables analyzed in the context of pain perception, as age, gender, and the prevalence of diabetes were included. We found no relationship between these predictors and the perception of pain in our osteoarthritis patients, so we did not report such results in this paper. As suggested by the reviewer, we improved the purpose of this paper and completed the appropriate conclusions in the discussion.

In addition, I would like to inform  that the authors of this study previously conducted comparative studies of the relationship between COMT and OPRM1 genotypes and pain sensitivity in healthy subjects. The obtained results did not confirm the relationship between the studied genotypes and pain sensitivity between professional athletes and the control group (Leźnicka K., Kurzawski M., BiaÅ‚ecka M., Safranow K., CiÄ™szczyk P., Malinowski D. Polymorphisms of catechol-O-methyltransferase (COMT rs4680:G>A) and μ-opioid receptor (OPRM1 rs1799971:A >G) in relation to pain perception in combat athletes. Biology of Sport. 2017;34:295-301 )

Secondly, all 207 patients were in the same group. Osteoarthritis has some stages, there are a lot of classifications, maybe the most popular is Kellgren--Lawrence scale. How can you be sure that there are no differences between patients who are in a different stage (or grade) of the disease? You should divide your participants in a few groups, according the stage of the disease. After measurements, you should compare results between subgroups, according to the stage or level of the osteoarthritis.

Authors:

In the limitations section, we have completed the description with the indicated problem.

We gave up to classify the osteoarthritis of evaluated patients, since we know from the literature that radiographic osteoarthritic severity is not a PROM (Patient-reported outcomes) predictor (Cho WJ, Bin SI, Kim JM, Lee BS, Sohn DW, Kwon YH, Total Knee Arthroplasty With Patellar Retention: The Severity of Patellofemoral Osteoarthritis Did Not Affect the Clinical and Radiographic Outcomes. J Arthroplasty. 2018 Jul;33(7):2136-2140. doi: 10.1016/j.arth.2018.02.075).

Reviewer 2 Report

Thank you for the opportunity to review this manuscript, which reports on an investigation of the association of COMT and OPRM1 genotypes with pain perception in patients after total ip replacement and total knee replacement. The review includes suggestions by section:

Title: suggest making the title more specific based on the findings, such as COMT polymorphisms associated with experimental pain in patients undergoing total hip or knee replacement.

Introduction: Provides context and justification for assessing variants of COMT and OPRM1 genotypes.

Page 2 Line 70 change "there is number" to "there are a number"

Materials and Methods: please clarify the time of data collection as preoperative or postoperative; otherwise all aspects are well explained.

Results: all analyses are well explained and visualization of data is appropriate.

Discussion: provides context of results within the wider literature; references are up to date. Page 14 line 82 replace "However, I many" with "However, in many". 

Limitations and Conclusion: appropriate

Thank you again for the opportunity to review the manuscript and my best wishes to the authors in their future endeavors.

Author Response

Dear Reviewer,

We would like to thank for careful reading our manuscript and for giving insightful comments, which helped us significantly improve our paper. All mistakes in manuscript have been corected.

The Authors

Reviewer 3 Report

In this manuscript, the authors conduct the genetic association between genetic variants of COMT and OPRM1 genes and pain perception among osteoarthritis patients after surgery of total hip replacement and total knee replacement. The authors included 207 patients and measured the level of pain. The authors reported that COMT genetic variants were associated with pain threshold but not pain tolerance. My comments are below.

·         I would recommend changing the title to present the gene name such as “genetic association between genetic variants of COMT and OPRM1 genes and pain perception among osteoarthritis patients after orthpedic surgery”.

·         In the abstract, I would recommend stating the genetic influence on pain perception in the background section, such as “experimental pain is highly variable by individuals, which is partly due to the genetics”.

·         In the abstract Material and Methods section, please describe which statistical analysis method was used in the study.

·         In the abstract Results section, please describe the result with numbers. Please state p-value, effect estimate (beta or OR and its 95% confidence interval) for the tested association.

·         In the Introduction, the logic of paragraphs is weak. The first, fifth, sixth, seventh and eight paragraphs contain most relevant information for this study, while the other paragraphs are less informative. I would recommend revising and reducing the Introduction section, focusing on 1st, 5th, 6th, 7th and 8th paragraph.

·         In 2.2. Participants section, please move the sentences in line 111-114 to the result section. For the sentence in line 114-115, please explain the eligibility criteria more in detail. Please describe how many patients were recruited in the initial stage and how many subjects were excluded by the eligibility criteria.

·         In 2.2. Participants section, please describe the ethnicity of the participants. As this study conducts the genetic association study, it is critical to describe it in the manuscript.

·         In 2.5. Genotyping section, please clarify the sentence in line 142-145, which is not clear. I would recommend clearly stating which SNPs for OPRM1 and COMT1 genes were genotyped, which will be helpful for the readers to get informed about how many SNPs were considered in the statistical analysis.

·         In 2.5. Statistical Analysis, I would recommend adding more methods based on the following comments.

1) Please describe how quality control for genotyped SNPs was conducted including not only Hardy-Weinberg equilibrium (HWE) but also missingness for SNP and subjects, sex mismatch and checking genetic ancestry.

2) Genetic ancestry is the most common confounder in the genetic association study. Do the authors perform the principal component analysis to check the genetic ancestry of the study population and adjust for the genetic ancestry in the statistical model? If not, I highly recommend conducting principal component analysis (researchers often use ‘smartpca’ software for this analysis) and including the first few principal components into the statistical model as covariates.

3) Please describe the p-value threshold for HWE test. As this study includes OA patients, I would recommend more stringent p-value threshold.

4) Based on Table 4, it seems that general (three genotypes) model indicates additive genetic model. Please change it to “additive”.

5) I would recommend specifying the outcome for the analysis and which statistical model was used, such as logistic regression or linear regression model.

6) Please describe which covariates were included in the analysis. As mentioned above, the principal components should be included to adjust for the genetic ancestry.

7) I would suggest applying the bonferroni corrected p-value not the nominal significance level (P=0.05) for the statistical significance threshold. As the authors tested multiple SNPs with multiple outcomes (5 SNPs * 8 outcomes = 40 testing), multiple testing issue presents and p-value needs to be more stringent.

·         I would suggest mentioning Table 2 before Table 1. It is conventional to present the demographics of the study population in Table 1 in the epidemiologic articles.

·         In Table 1, please change ‘alleles’ to ‘major/minor’ allele. Please present minor allele frequency not major allele frequency. ‘Missing’ column is confusing. Is it missingness for SNP or missingness for subject? Please clarify it. Please add the chromosomal position (chromosome and base position). Please add the functional consequence for each variant.

·         In Table 2, ‘Mean +/- SD, N(%)’ header is confusing. I would recommend dropping the last 5 rows, which is the count of each genotype. This information is just equivalent to the minor allele frequency.

·         In Figure 1, please add the name of y-axis. Please spell out “ProgHP”.

·         The author presented multiple Tables by each variant, which makes it difficult for the reader to understand the main finding in this study. I would suggest collapsing Table 3, 4, 5, 6 and 7 and presenting it as a supplemental table. In the main manuscript, I would suggest presenting a table with nominally significant results (P<0.05). Please change “General” to “Additive”. Please add the legend to state which was the effective allele in the analysis. For example, in Table 4, the effective allele is T allele.

·         The result shows that SNPs, rs4633, rs4680, rs4818 and rs6269, are associated with pain threshold (RH). I assume that it is due to they are highly in linkage disequilibrium (LD). It would be helpful to include the LD block plot of those 4 variants as a supplementary figure.

·         In Discussion, the sentence in line 34-37 is not correct. When conducting the association between exposure and outcome in the regression model, we seek to find whether exposure tends increase or decrease the risk of outcome on average or not. It is not true that all the individuals with the carriers have less pain than those without the carriers. The regression model estimate the average value so it should be interpreted that those who have homozygous genotypes for AA of COMT genetic variant have on average less pain than those who have not. Please correct accordingly.

·         The author investigated the genetic susceptibility of pain after the surgery among OA patients. It is likely that pain acceptance and OA risk partly share the genetics. Do the authors look up genetic variants tested in this study to the most recent genome-wide association study of OA (Boer, Cindy G., et al. "Deciphering osteoarthritis genetics across 826,690 individuals from 9 populations." Cell 184.18 (2021): 4784-4818.)? It would be helpful to describe the previous finding of OA genetics for the variants tested in this study in the Discussion section.

·         In Conclusion section, the sentence “This association was observed in patients with two AA alleles (rs4680 and rs6269), TT alleles (rs6633)  and CC allels (rs 4818) of the COMT gene, who had significantly higher pain thresholds on the right hand.” is wrong. It should be stated such as “in our study population, A allele of rs4680, COMT gene, is associated with higher pain thresholds on the right hand”.

Throughout the manuscript, typos and grammatical errors are present. The English editing is highly recommended.

Author Response

Dear Reviewer,

We would like to thank you for careful reading our manuscript as well as pertinent, substantive comments, which helped us significantly improve our paper. We have revised the manuscript and responded to the comments of the reviewer.

The authors

Round 2

Reviewer 1 Report

1. I'm not satisfied with your answer about study design. My complaint stay the same still. You have proved some connections between COMT and OPRM1 genotypes and pain perception in patients with osteoarthritis of the hip and knee. But maybe you have the same situation in a cohort without osteoarthritis. You should compare two cohorts (with and without osteoarthritis) and than make a conclusion. I found your paper where you had professional athlets and the control group. In that paper you had two groups of healthy people, because of that you can't aply or use the same conclusion in paper under review.

2. On the basis of only one published paper, it is not possible to say that radiographic osteoarthritis severity is not a PROM  predictor. 

Author Response

Dear Reviewer,

1.The aim of our project was to describe the relationship of polymorphisms of selected genes with the perception of pain in sick people, taking into account such predictors as: age, back pain, diabetes (as well as hypertension - not shown at work), which significantly affect the perception of pain in healthy people. It was not our goal to compare patients qualified by specialist doctors for hip and knee arthroplasty with healthy people. This is not possible in this project.

2. The publication sent is one of many that indicate that the radiological severity of osteoarthritis does not have to correlate with the complaints reported by the patient.

Kind regards,

The Autors

Reviewer 3 Report

1. In the abstract Material and Methods section, please specify which statistical analysis model was used, not software name.

2. In the abstract Results section, please revise this sentence “In the study group, we found no statistically significant differences between the COMT and OPRM1 genotype distributions and pain thresholds and pain tolerance” to “In the study group, we did not find statistically significant genetic associations between variants of COMT and OPRM1 and pain thresholds/tolerance”.

3. In the abstract, the last sentence is wrong. Please revise the sentence such as “Moreover, any genetic variants were not associated with subjective pain perception using the VAS scale”. Also, please spell out VAS.

4. In regards to genetic ancestry and principal component analysis (PCA), although Polish population is relatively homogeneous, it is possible that genetic variants of interest are not homogeneous in the population. Such cryptic substructure must be considered by performing PCA and the statistical model should include PC components to adjust for the genetic ancestry. Given the author’s decision not to perform PCA, I highly recommend stating the limitation that this study did not consider the genetic ancestry in the model, which may result in the confounding for the association.

5. The authors did not expand 2.6. Statistical Analysis section. The authors pointed out what they had stated in the previous manuscript. 

Round 3

Reviewer 1 Report

Dear Authors,

Thank you for your response. 

Author Response

Thanks for your comments.